# Bee Hotels as a Tool for Post-Fire Recovery of Cavity-Nesting Native Bees

**DOI:** 10.3390/insects16070659

**Published:** 2025-06-25

**Authors:** Kit Stasia Prendergast, Rachele S. Wilson

**Affiliations:** 1School of Molecular & Life Sciences, Curtin University, Kent St, Perth, WA 6102, Australia; 2Institute of Life Sciences & the Environment, University of Southern Queensland, 487-535 West St, Toowoomba, QLD 4350, Australia; 3School of Biological Sciences, The University of Queensland, University Drive, St Lucia, QLD 4067, Australia; 4School of BioSciences, The University of Melbourne, Royal Pde, Melbourne, VIC 3052, Australia; 5Centre for Planetary Health and Food Security, Griffith University, Kessels Road, Brisbane, QLD 4111, Australia

**Keywords:** bees, conservation, fire, native bees, post-fire, restoration, trap nests, wild bees

## Abstract

Wildfires are increasing in extent and severity under anthropogenic influences, with potential adverse impacts on native pollinators like wild bees. Whilst there has been much attention paid to assessing the impact of wildfires on vertebrates and assisting in their recovery by augmenting habitat that has been destroyed, there has been limited research into assisting invertebrates. Here, we use bee hotels to assess their ability to support the recolonisation of natural habitats after the 2019/2020 Australian bushfires. One thousand bee hotels were installed across five sites in the Jarrah forest of southwest Western Australia and monitored from August until March. Uptake was slow, but by March, every bee hotel had at least one nest, and some had all nests occupied. Over 900 nests were occupied, and there was a greater number of native bees observed foraging at sites where bee hotels were installed compared with three burnt sites where they were not. We also observed a significant negative relationship between honeybee density and nesting occupancy and native bees foraging, suggesting this introduced species can outcompete native bees in post-fire habitats. Overall, we demonstrate a proof-of-concept that bee hotels can assist in the recovery and monitoring of cavity-nesting native bees in post-fire landscapes.

## 1. Introduction

Wildfires are increasing in prevalence and intensity under anthropogenic climate change and pose threats to plant-pollinator communities [1,2,3,4]. Much of the Australian landscape is fire-prone, and whilst fire has been part of some habitat types, the extent and nature of recent fires are unprecedented [5,6].

In the summer of 2019/2020, approximately 10.3 million hectares of Australia were burned, threatening wildlife and ecosystems [6,7,8]. Fire can promote regrowth or stimulate flowering, thus accruing benefits to bees from the flush of food resources [9,10]; however, bees also need to nest. With limited flight ranges of 50 m–3 km [11,12], a lack of nesting resources could limit any benefits from flowers in post-burn areas and prevent recolonisation. For below-ground nesting bees, fire can benefit this guild by opening up the canopy and providing patches of bare ground [13,14,15]. However, for above-ground cavity-nesting bees, fires may remove their nesting resources. In severe fires, trees are destroyed or rendered unusable for nesting [16,17]. It can take decades for trees to reach maturity and for new wood-borer holes to be available. Hence, we can predict that nests may be a limiting factor for cavity-nesting bees in post-fire habitats. Whilst there has been research on pollinator responses to fire, primarily in North American temperate forests and Mediterranean ecosystems [4,18,19], how cavity-nesting native bee assemblages respond to fire has not been investigated in Australia, and importantly, there has been no investigation into practical solutions to aid in recolonisation and recovery of cavity-nesting bees that utilise pre-made cavities.

Cavity-nesting bees are known to be affected by fires, from research conducted in other countries, owing to loss of nesting substrates [19,20,21]. Research in Australia of *Xylocopa (Lestis) aerata*, a carpenter bee that makes its own cavity nests, has also revealed that fires can have devastating effects on their populations [22]. Cavity-nesting bees that are “renters” do not excavate their own nests, but instead rely on small holes of appropriate diameters to be created and then vacated by wood-boring beetles [23]. We can expect that a lack of nesting resources may be most acute for the cavity-nesting guild.

Bee hotels (or trap nests) are artificial nesting structures that provide cavity-nesting bees with increased nesting substrate opportunities [24] and may also be used by other cavity-nesting insects like wasps [25]. In Australia, bee hotels have been shown to be used with success as a tool for monitoring populations [26,27,28,29,30]. They have been used effectively to date to boost native bee populations in agricultural landscapes [31]. Other artificial nesting substrates have been used to monitor and support the recolonisation of “carpenter” bees (*Exoneura* and *Xylocopa* species) populations post-fire” [22,32]. However, they have yet to be evaluated as a tool for aiding cavity-nesting bee colonisation of burnt landscapes.

In recently burnt landscapes, foraging resources may also be limited, especially for flowering resources where the plant species can take decades to reach maturity (e.g., [15,33,34,35]). Under conditions of limited resources, it can be expected that interspecific competition will be higher and lead to detrimental impacts on populations of species that are inferior competitors [36]. One highly competitive species is the introduced European honeybee *Apis mellifera*. This super-generalist, eusocial invasive bee species has the potential to have detrimental impacts on native bees, especially those that have high niche overlap and when those floral resources are relatively less common [37,38,39].

Our study is the first to trial bee hotels as a post-fire recovery option. Nesting habitat is a vital part of the equation for recovering pollinators, for even if they can move into fire-affected landscapes and forage on flowering regrowth, they will be unable to establish without nesting resources. Our study also aimed to investigate which bee hotel design type (bamboo vs. wooden) was more successful in promoting occupancy, and if honeybee density was associated with reduced native bee densities in post-fire habitats.

## 2. Materials and Methods

### 2.1. Study Sites

The project was conducted in regions of high biodiversity conservation, namely at sites of high conservation importance (state forest and reserves) in the southwest Western Australia biodiversity hotspot [40] (Figure 1B). Sites were located where 2019/2020 bushfires had occurred in the Northern Jarrah Forest, based on the AUS GEEBAM Fire Severity Dataset (https://fed.dcceew.gov.au/maps/aus-geebam-fire-severity-dataset-2019-2020/about, accessed on 14 November 2020). The vegetation is classified as sclerophyllous, broad-leaved evergreen, open canopy forest, and forests of the southwest are dominated by a few key tree species, including jarrah (*Eucalyptus marginata*), and to a lesser extent, marri (*Corymbia calophylla*) [41,42]. There is a diverse understorey, with over 850 described species of vascular plants [43]. Dominant mid-storey plants include *Banksia grandis* Willd., *Allocasuarina fraseriana* Miq. L.A.S.Johnson, and *Persoonia longifolia* R.Br [43]. The region has a Mediterranean climate-type ecosystem with hot, dry summers, and cool, wet winters, yet for over 45 years has been experiencing rainfall decline and temperature increase [44]. The region has also been subject to logging and bauxite mining [45].

Prior to European colonisation, the Jarrah forest had very low incidences of fire injury, with the average interval between injurious fires being about 81 years. Following European colonisation, fire incidence increased in frequency and severity up until the 1950s, owing to anthropogenic activities leading to greater fuel levels (i.e., from logging) [46]. Fires have since increased due to climate change, as well as fuel-reduction burns (prescribed burns), with recent fire management aiming for stands to be prescribed burned at low severity on 5–12 year intervals [47,48]. The risk of severe fires is increasing, as southwest WA has experienced a significant reduction in annual rainfall since 1975 [49], whilst fire weather conditions suited for wildfires have increased [50], with drastic impacts on biodiversity [51].

The study was conducted in August–March 2021/2022, and as the sites were burned in the summer of 2019/2020, the sites represented habitats two years post-fire. Each site was at least 2 km from the rest to ensure independence (beyond the foraging range of most native bees [52]). Bee hotels were placed in burnt areas approx. 15 m away from unburnt areas to avoid edge effects but also to enable colonisation from the unburnt patches.

### 2.2. Bee Hotels

At each site, 20 bee hotels of two design types (10 wooden and 10 bamboo) were installed at five sites in three regions that were impacted by the 2019/2020 bushfires (Figure 1A). Wooden bee hotels were Jarrah (*Eucalyptus marginata*) untreated wooden blocks 20 cm deep, with 15 holes of three diameters each (4 mm, 7 mm, and 10 mm) and 15 cm long (Figure 1B). Occupation by bee-hotel hole diameter was assessed to determine any preferences. Bee hotels followed the same design as those used in other studies of cavity-nesting bees [53,54], see also [26]. In addition, 10 bamboo bee hotels with approx. 50 bamboo canes of diverse diameters (Figure 1C), 15 cm long in PVC pipe, were also installed. A pair of bee hotels (one of each material) was hung up together at approx. 1.5 height on a tree branch. Bee hotels were installed at each site across an area of approximately 500 m^2^.

Bee hotels were installed in August, and surveys were conducted monthly during bee activity season (installed in August 2021, monitored September 2021 until March 2022) to monitor occupancy. Bee hotel monitoring was conducted by a single native bee scientist (KP), therefore preventing inter-observer bias. Each bee hotel was numbered with a permanent marker to allow tracking of occupancy over time. During each visit, each bee hotel was photographed with an iPhoneX, and the number of nesting holes occupied was recorded. For the wooden bee hotels, the diameter of the occupied holes was also recorded. The taxon of the insects using the hotels was recorded, and, if the occupant was a native bee, the genus (*Megachile*, *Hylaeus*, or *Exoneura*) based on the type of nest cap. If a native bee was observed at her nest during the bee hotel monitoring, the particular species was noted. These were identified by KSP based on her previous native bee biodiversity surveys in the region, where insects were collected and identified using relevant taxonomic keys, referencing the WA Museum Entomology Collection and online databases such as PaDIL.

### 2.3. Surveys of Bees Foraging on Flowers

During each visit to check the hotels, two hours were spent by KSP observing flowering plants and recording the number of bee visitors to the plants and the taxon of each visitor. These surveys covered an area of approx. 700 m^2^ in and around the area where the bee hotels were installed. Bees were identified to genus; specimens were not collected to ensure that, during this vulnerable post-fire recolonisation period, there would be no reductions in bee populations or reductions in females that could potentially use the nests. Flowering plants were identified to species-level with field guides and by cross-checking photographs with botanists in the region (K. Dixon). These surveys served to (a) identify important food sources for native bees in post-fire habitats; (b) determine if ground-nesting bees as well as cavity-nesting bees were present; (c) determine if cavity-nesting taxa were present in the area, thereby enabling us to gauge the effectiveness of the bee hotels, distinguishing between not being used vs. the cavity-nesting species not being in the area; and (d) provide an estimation of whether their numbers were increasing following the installation of bee hotels. Three fire-affected sites that served as controls where no bee hotels were installed were also surveyed for foraging bees for two hours each across an equivalent area (700 m^2^), thereby enabling a comparison as to whether bee hotels significantly boosted cavity-nesting bee populations. These sites were located greater than 5 km away from the sites where bee hotels were installed (beyond the foraging distance of native bees, and beyond the average foraging distance of honeybees, ensuring independence), but no greater than 20 km away. They were selected to be in the same landscape which was burned during the same time as the burnt sites in which bee hotels were installed, ensuring that any differences were likely due to the effect of the bee hotels, rather than to biogeographic differences, for example, that could confound the effects of bee hotel presence/absence.

During these visual surveys, honeybee density (number of honeybees visiting flowers) was also recorded to investigate the potential for competition between honeybees and native bees in recently burnt sites.

### 2.4. Data Analysis

Statistical analyses were performed in R version 4.2.1 [55]. Data were analysed using generalised linear mixed-effect models (package “lme4” [56]). A Poisson distribution was used, and model fit was assessed with the package “DHARMa” [57]. Site was included as a random factor. Significance (at *p* < 0.05) of the fixed effect variable of interest was assessed using a log-likelihood approach, performing an ANOVA between models with and without the variable of interest. Estimates were calculated from a summary of the model.

Generalised linear mixed-effect models were used to evaluate the hypothesis that there would be a greater number of native bees observed at the treatment (bee hotel sites) than control sites. As bee hotels are predicted to only assist in the recovery of cavity-nesting rather than ground-nesting bees, each guild was analysed separately. Due to overdispersion, an observation level as a random factor was included. GLMMs were also used to test the association between the abundance of honeybees observed with bee hotel occupancy and native bees observed foraging.

GLMMs were composed using the lme4 package with the fixed factors of number of native bees foraging (or honeybees foraging) and treatment (bee hotels installed vs. no bee hotels), and the random factors of site and month (lme4 package). An observation-level random effect was added to models that were overdispersed [58]. Models were compared against null models (i.e., random factors only) to assess the significance levels of explanatory factors with likelihood ratio tests (χ^2^) using the *anova*() function [59]. The explanatory power of the models was determined from the variance of fixed (marginal *R*^2^) and random effects (conditional *R*^2^) with the MuMIn package [60]. Differences between levels of categorical explanatory factors were compared using Tukey’s post hoc tests with the *glht*() function [61].

## 3. Results

### 3.1. Bee Hotel Colonisation

Bamboo and wooden bee hotels (Figure 1B,C) were installed just before spring (in August) and were occupied by bees from October 2021. By the end of the study period (March 2022), the majority of bee hotels were occupied at every site (Appendix A). The total number of cavities occupied was high, with up to 283 total nests at just one site (Appendix A). Across all bee hotels in all five sites, a total of 832 cavities were occupied by native bees (Appendix A).

There was some variation in bee hotel uptake between the sites. The Jarrahdale locations had very high occupancy (maximum occupancy by December at one site and March at both sites; Appendix A) and also had the highest number of nests occupied in total by the end of the project (Figure 2A). Even the site with the slowest uptake, Wellington State Forest, had seven to nine (out of ten) wooden and bamboo bee hotels occupied, respectively, by March (Appendix A).

No bees were observed using bee hotels across all sites until October. The major increase in occupancy occurred in January and March at Jarrahdale, and March for Harris River, especially at one of the sites (Figure 2B).

### 3.2. Bee Hotel Material

Every one of the wooden and bamboo bee hotels was used by native bees (Appendix A). In terms of the total number of nests, 573 bamboo nests were occupied (average across the five sites: 114.6 ± 22.2), and 327 wooden nests were used (average: 65.4 ± 19.7) (Figure 2B). However, there were a greater number of available nests per bamboo hotel (approx. 50 for bamboo bee hotels, vs. exactly 15 in the wooden bee hotels). In terms of the proportion of holes available, by the end of the study, an average of 0.23 ± 0.04 bamboo nests were occupied, compared with an average of 0.44 ± 0.13 wooden nests (Figure 2C). For the wooden bee hotels, the most frequently occupied hole diameters were 7 mm (136 nests), followed by 4 mm (113 nests), then 10 mm (78 nests) (Appendix A).

### 3.3. Foraging Bee Surveys

Ground-nesting bees were observed during the earlier months, but as the season progressed, cavity-nesting bees dominated the observed bees (Appendix A). More bees were observed foraging in the field at the sites with bee hotels (mean = 41.3 ± 28.5 per day per month) compared to the sites without bee hotels (mean = 1.62 ± 0.75; delta AICc = 10.4, *p* < 0.01) (Table 1). Thus, there were significantly more native bees foraging in sites with bee hotels (IQR = 0.5–15.5) than without (IQR = 0–4; χ^2^ = 9.68, *p* = 0.002; Figure 3A) (Table 1). The introduced *Apis mellifera*, on the other hand, were found in similar numbers at treatment (IQR = 0–3) and control sites (IQR = 0–1; *p* > 0.05; Figure 3B) (Table 1).

### 3.4. Taxonomic Composition of Bee Hotel Communities

*Megachile* was the most frequent occupant (88.7% of nests overall), as evidenced by the resin, resin and sand, or chewed plant material nest caps (Table 2). Of nests occupied by megachilids, 65% were in bamboo hotels and 35% in wooden hotels (Table 2). No megachilids in the *Eutricharaea* subgenus (“leaf-cutters”) used the bee hotels (Appendix A). Hylaeinae, which cap their nests with a polyester-like secretion, were the next most common bee occupants (2.4% of all hotels), with a higher proportion of wooden hotels being occupied (59% of Hylaeinae nests) compared with bamboo hotels (20.9%) (Table 2). *Exoneura* bees used the 4 mm wooden nests as well as a few observations in bamboo stems (Table 2).

Although many native bee species cannot be identified based on field observation alone (as diagnostic characteristics are microscopic), the megachilids observed using bee hotels during bee hotel monitoring (*Megachile erythropyga*, *M. aurifrons*, *M. oblonga*, and *M. monstrosa*) are common species and could be identified confidently.

A variety of non-target organisms were found in the bee hotels (Table 2), especially during the earlier months (Appendix A). These were mainly spiders and ants, but also wasps and crickets (Table 2). Even by the end of the project, Formicidae occupied almost 5% of occupied nests, with equal numbers of bamboo and wooden nests being occupied.

### 3.5. Foraging Resources

Regenerating flower resources used by bees were identified in field surveys (Appendix A). No exotic flowers were visited by native bees, whereas some exotic “weeds” were visited by *Apis mellifera*. Earlier in the season (spring), native peas (Fabaceae: *Bossiaea*) were important foraging resources, and later in the season (summer), *Corymbia calophylla* (Myrtaceae) trees were an important resource.

### 3.6. Honeybee Competition

We found that fewer native bees were observed and fewer bee hotel cavities were occupied if honeybee numbers in the area increased. There was a significant negative association between bee hotel occupancy and the number of honeybees observed (estimate −0.451, se 0.022, delta AIC 1122, *p* < 0.01) (Figure 4). There was also a significant, but weak, negative association between honeybees observed and native bees observed (estimate −0.157, se 0.011, delta AIC 344, *p* < 0.01).

## 4. Discussion

We have demonstrated that installing bee hotels as designed here in post-fire habitats of southwest Australia can aid recovery of cavity-nesting bees. Every bee hotels was occupied, and with a total of 832 artificial nesting holes provided by these custom-designed bee hotels, this represents a sizeable native bee population to colonise the post-fire recovering habitat. Assuming an average of four cells per nest [54], this means that from the 832 nests that were occupied, 3328 native bees would emerge in the next generation.

Both bee hotel materials were used across all sites. In terms of total holes occupied, there were just under twice as many bamboo nests used, but in terms of proportion of nests available, just under twice as many wooden nests were used. Unlike for the wooden bee hotels, the bamboo hole diameters could not be standardised, so it is unclear whether the lower proportional occupancy in bamboo hotels was due to a larger proportion of hole diameters that were less preferred by the cavity-nesting bee populations. Future studies that standardise the number of holes and approximate hole sizes available between the bamboo and wooden bee hotels will be useful to determine whether there is a preference for one particular bee hotel material over the other. This research, however, does indicate that both bee hotel types can be used, which provides citizen scientists with more flexibility in materials based on local availability in terms of making bee hotels using these design principles.

Surveys at control sites (nearby burnt sites with no bee hotels installed) had relatively few bees, especially during the latter part of the season, suggesting that sites with bee hotels were attracting bees to the area as the season progressed (Appendix A). The greater number of cavity-nesting bees at sites where bee hotels were installed, but not ground-nesting bees or honeybees, further supports the conclusion that bee hotels aided in attracting and establishing this guild (cavity-nesting bees). Due to the budget constraints that limited the duration of this project, we were unable to conduct longer-term follow-up monitoring; however, we recommend that future studies conduct research over subsequent years to determine if the provision of bee hotels in post-fire habitats increases the local cavity-nesting native bee population over generations.

This work fills a major critical gap, for whilst there has been much attention on flora recovery, there has been virtually no investment into the recovery of the pollinators of flora, which are absolutely vital for the sustainability of plant populations. This work will be important in providing nesting resources for cavity-nesting bees that are eliminated in burnt habitats, enabling the recolonisation and establishment of cavity-nesting bee populations.

The uptake of bee hotels only starting to progress in October, then substantially increasing over the summer, is consistent with previous research in southwest WA, where the main period of activity for cavity-nesting bees in this region is not until November–February (see [62,63]. Slow uptake at some sites may have been due to few flowering resources during the time when the cavity-nesting bees were active (pers obs.).

The generic composition of bees using bee hotels is similar to that from urban bee hotel studies in the same greater region of southwest Western Australia), However, unlike in the previous study [54], *Exoneura* were observed using 4 mm holes in bee hotels. Interestingly, other bee hotel studies across Australia [26,27,28,29,30,64] have not recorded Allodapini using bee hotels of this design, as this taxon typically prefers pithy stems/reeds [65]. Stem-nesting species are predicted to be the most nesting resource-limited after fire [66], as such substrates are most easily completely destroyed, even by low-burning and low-intensity fires. It may be that when their preferred nesting substrate is removed, they will use these alternative above-ground nesting resources.

The only study to date in Australia involving bee hotels and fire is that by Gilpin, Brettell, Cook and Power [64]. However, they did not install bee hotels in areas that had been burned as a recovery tool; rather, the bee hotels were installed prior to fire. The authors found no native bee occupancy after fire, but also no occupancy by bees even prior to fire. Discrepancies may relate to the poor-quality habitat for bees in the agricultural landscapes in which that study was conducted.

Whilst bee hotels have not been deployed as recovery tools, artificial nesting substrate in the form of balsa wood posts for *Xylocopa aerata* bees has been deployed as a recovery tool after devastating fires eliminated their natural nesting habitat on Kangaroo Island. Here, it was similarly found that augmenting the nesting substrate aided in the recovery of this species [22].

A desktop assessment and modelling of native bees in Australia in relation to the 2019/2020 bushfires revealed that thirteen above-ground nesting species would be eligible for threatened-species listing according to IUCN Red List Criteria [66]. Unfortunately, like many native bee species in Australia, their preferred nesting substrates are unknown. Research into hole diameter, height, and preferred wood type of these species will allow targeted bee hotels to be made to assist the populations of these vulnerable species.

We have demonstrated that providing nesting substrate that is vital for the establishment and recovery of cavity-nesting bees, these allowed these bees to successfully colonise areas, and enhance their populations provided that the opportunity for their populations is enhanced. This is expected to have flow-on benefits, for their populations are important for the pollination of wildflowers [67,68,69]; without pollinators, any flora that are dependent on these pollinators that survive or germinate in the fire-affected areas will fail to set seed. Therefore, these recovery actions are important for the persistence of both bee and plant populations.

As bee hotels can be reused, their benefits can be ongoing, allowing populations to persist until natural nesting substrate becomes available. Maintenance is low, and park rangers, including indigenous rangers, can maintain the bee hotels over the long term. We do recommend that nests from which bees have emerged be cleaned out (e.g., with a damp pipe cleaner) to remove mites or disease before the next generation reuses the nests, thereby reducing parasitism [70].

We were also able to identify which floral resources were utilised in a post-fire environment. The exclusive visitation by native bees to native flora, vs. the visitation by honeybees to also exotics, is consistent with the preference of native Australian bees for Australian flora compared with the more generalist flower preferences of the exotic honeybee [39,71]. It should be noted that, during some surveys, almost no flowers (exotic or native) were observed.

Whilst the native *Bossiaea* is considered a “leguminous fire-weed” [72], the *Corymbia calophylla* was able to flower because the trees had survived, and the canopy was flowering. However, with a devastating fire, mature trees may not survive, and regrowth could take at least half a decade to reach maturity [73]. The visitation to *C. calophylla* reinforces the value of this flowering tree in late summer/early Autumn, when almost no other plants are in flower [74]. This species is a resprouter, so mature trees are able to flower after fire, but if the fires are so intense as to cause mortality of these trees [17], there would have been a major nutritional resource shortage for the bees. Our results reinforce the need to protect large flowering trees and adopt fire management practices that preserve such resources in the environment.

Honeybees have the potential to compete with native bees for flowering resources (nectar and pollen), especially if resources are limited [38], such as post-fire, which can impair the fitness of cavity-nesting native bees [75]. Indeed, we found evidence that honeybees may be negatively impacting native bees in post-fire environments. This suggests that in areas affected by fire, beekeeping should be excluded so as to allow native bees to recolonise or persist in the post-fire environments.

### Future Research Recommendations

Due to time constraints, we were unable to monitor the hotels to determine if progeny emerged. However, previous research using these same designs found that offspring successfully developed and emerged. Future research, however, is required to confirm that progeny survive and emerge from bee hotels in post-fire landscapes, because if resources are limited, offspring emergence may be relatively low.

A comparison of pre- and post-fire cavity-nesting bee populations at the same sites would be ideal to determine the impact of fire and recovery actions. Such baseline data is unavailable; however, surveys of foraging bees in sites with and without bee hotels found that there were significantly more bees observed in the sites with bee hotels. Furthermore, a previous study of bee hotel communities in urban sites in southwest Western Australia in 2016/2017 and 2017/2018 [54] found a lower occupancy among bee hotels (13.3% nests occupied in 2016/17 and 6.34% nests occupied) compared to the present study (approximately 23% of bamboo nests and 44% of wooden nests occupied). This suggests that suitable nesting cavities are limited in burnt habitats and that installing bee hotels can aid in recolonisation by native cavity-nesting bees.

Ideally, we would have liked to have installed bee hotels at appropriate control sites where fires had not occurred. This would have allowed us to better test the hypothesis that nest sites are limiting. However, due to the constraints of the Bushfire Recovery Grant, this was outside the scope of the grant guidelines. We recommend that future studies, where possible (acknowledging the difficulty of such studies when evaluating wildfire responses by wild animals), conduct a before–after study with replicated sites representing burnt vs. unburnt treatments, with and without bee hotels, for a more rigorous test of the hypothesis that bee hotels can aid in post-fire recovery. We also recommend that future research evaluate the availability of floral resources used by bees in these different treatments, as measuring such variables was not possible with the person power of this research (one person).

## 5. Conclusions

Our on-the-ground intensive bee hotel installations and monitoring demonstrate that providing bee hotels for cavity-nesting bees allows native bees to recolonise post-fire habitats. Comparing native cavity-nesting bee populations and bee hotel occupancy at unburnt sites in the same region could further indicate the impact of fires on this guild of native bees and the use of bee hotels in burnt areas.

## Figures and Tables

**Figure 1 insects-16-00659-f001:**
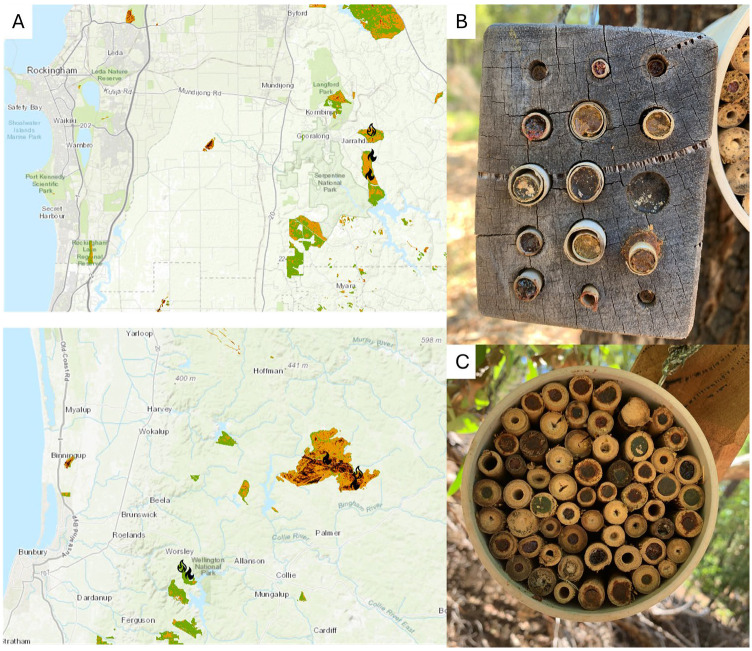
Map (**A**) showing locations of bee hotels installed in Western Australia (infilled fire symbol) and control areas (outlined fire symbol) in Western Australia. Map shows areas burned by bushfires in 2019/2020, with fire severity from lowest (green) to highest (maroon). Fire map from The Australian Google Earth Engine Burnt Area Map (AUS GEEBAM): a rapid, national approach to fire severity mapping. Fire severity is defined as a metric of the loss or change in organic matter caused by fire using a vegetation index (Relativised Normalised Burnt Ratio, RNBR) that is calculated for burnt areas and adjacent unburnt areas, before and after the fire season. The result is a map of four fire severity classes that represent how severely vegetation was burned during the 2019/2020 fires. Top map shows the Jarrahdale sites; bottom map shows the Harris River and Wellington State Forest sites. Bee hotels installed in pairs at each site included wooden (**B**) and bamboo (**C**) types. Occupancy is “capped” in nests.

**Figure 2 insects-16-00659-f002:**
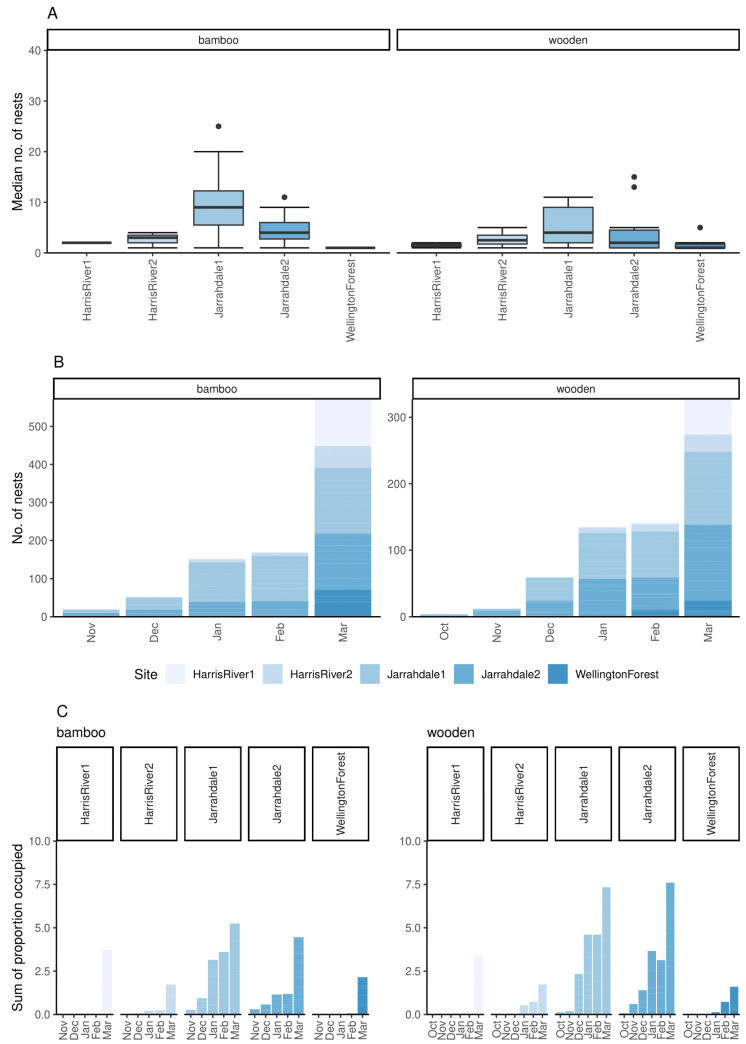
Occupation of bamboo (left) and wooden (right) substrates by native bees from September 2021 to March 2022: (**A**) Box and whisker plot showing median and first and third quartiles (box) and range (whiskers) of the number of cavities occupied by native bees at the end of the study; (**B**) cumulative number of cavities occupied each month across all sites; and (**C**) cumulative proportion of cavities occupied each month per site. Number of provided bee hotels per site per substrate = 10.

**Figure 3 insects-16-00659-f003:**
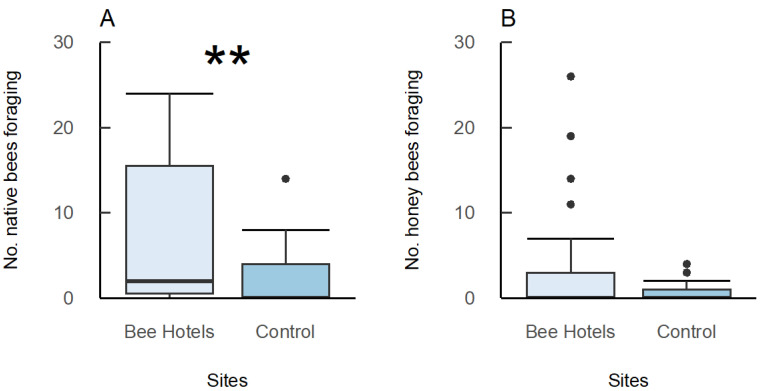
Number of bees observed foraging on average per survey at treatment sites (with bee hotels; 35 surveys) and control sites (without bee hotels; 21 surveys) for native bees (**A**) and honeybees (**B**). Asterisks denote significant differences in bee numbers between treatment and control sites (** = *p* < 0.01).

**Figure 4 insects-16-00659-f004:**
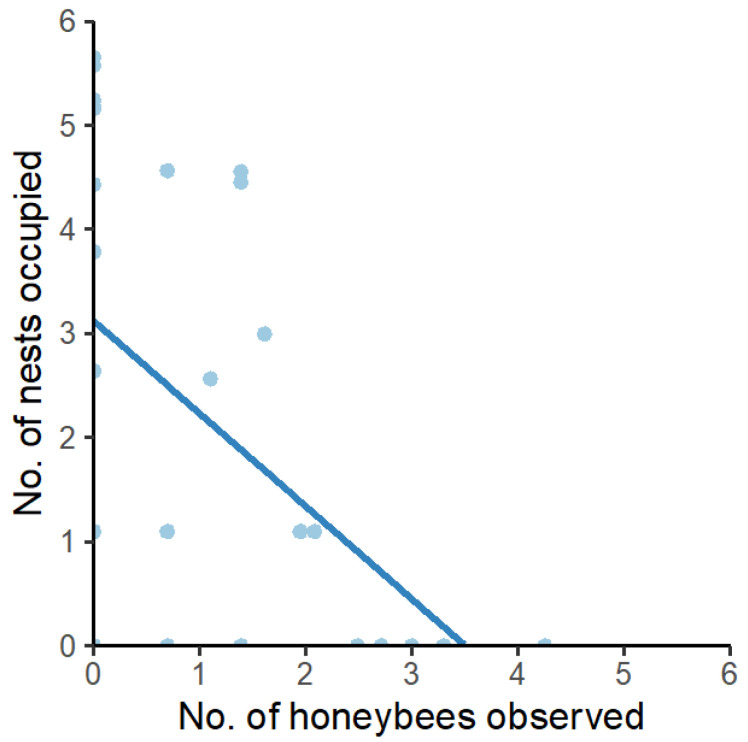
Scatterplot of the number of nests occupied in bee hotels and the number of honeybees observed foraging in the same sites (blue points). Data are log + 1 transformed for visualisation purposes. Dark blue line indicates negative association between variables (estimate = −0.451 ± 0.022; *p* < 0.01).

**Table 1 insects-16-00659-t001:** Parameters and statistics for GLMMs comparing native bees and honeybees observed foraging at sites with and without bee hotels. Note that all GLMMs used the random factors of site, month, and an observation-level effect. The explanatory power of the most parsimonious GLMMs, including statistically significant variables, is shown as a variance of fixed (marginal R^2^) and random effects (conditional R^2^). Significance levels of variables were assessed by likelihood-ratio tests (LRT) against null models (i.e., random factors only). *p*-values of less than 0.05 are indicated in **bold**.

Model	∆ R^2^		LRT			Tukey’s Post-Hoc	
	Marginal	Conditional	χ^2^	df	*p*	Levels (Direction)	*p*
N native bees foraging~Treatment	0.2013	0.9954	9.683	1	0.0018	Bee hotels > Control	**0.0023**
N honeybees foraging~Treatment	0.0567	0.9614	1.396	1	0.2337	Bee hotels > Control	0.2000

**Table 2 insects-16-00659-t002:** Taxa observed in bee hotels by substrate at the completion of bee hotel monitoring (March). Data are shown as the number of nests occupied and their relative proportions. Note that earlier in the project, prior to native bees entering, some holes were temporarily occupied by other taxa, including wasps (Eumeninae, as well as *Isodontia*), Araneae, Formicidae, a Diptera, two Orthoptera, and a Scincidae (Appendix A).

Group	Taxon	Bamboo	Wooden	Total	% Nests
Native bees	Megachilidae	521	285	806	88.67
	Hylaeinae	9	13	22	2.42
	*Exoneura*	2	2	4	0.44
Wasps	Eumeninae	8	4	12	1.32
Ants	Formicidae	28	11	39	4.29
Spiders	Aranae	13	13	26	2.86

## Data Availability

The original contributions presented in this study are included in the article/Appendix A. Further inquiries can be directed to the corresponding author.

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
