# Peer review of "Bee Hotels as a Tool for Post-Fire Recovery of Cavity-Nesting Native Bees"

_insects, 2025, doi:10.3390/insects16070659_

Round 1

Reviewer 1 Report

Comments and Suggestions for Authors

Overall, this is a relevant study and one that addressing a growing concern related to climate change, its consequences (increases in wildfire), and the impacts on native bee populations. The study design is interesting, and the authors outline the pros and cons as well as potential pitfalls and future directions appropriately. Analytical methods are sufficient and clearly described. There are a few main considerations that should be addressed prior to publication. The first is a lack of reference to the broad series of literature of on the effects of fire on pollinator communities globally. Much work has been done in North America (and elsewhere) in both forest and grassland systems. There is room for this topic to be referenced however the authors choose in the introduction - especially the first paragraph (see specific comments below). The second main point of emphasis regards Figure 2A - this figure is a bit confusing (the caption related to 2A does not clearly describe what is shown) and I have provided some suggestions about how to make this figure clearer and more impactful. Lastly, I would suggest a careful read through the manuscript because I found several instances where either extra words were included in sentences or words were left out (some examples in specific comments below). 

Line 67: You state here: “Despite this, how native bee populations respond to fire, and practical solutions to aid in recolonization and recovery, have not been investigated. “

You already cite this reference in a previous sentence: Carbone, L.M., Tavella, J., Pausas, J.G. and Aguilar, R., 2019. A global synthesis of fire effects on pollinators. Global Ecology and Biogeography28(10), pp.1487-1498.

And that reference undermines your statement that native bee responses to fire has not been investigated.

There is a robust series of literature on native bee responses to fire in other systems globally that should be noted and can bolster the predictions presented here. I would suggest the inclusion of at least some citations of these studies and a reframing of this sentence if meant to state that this topic has not been studied in Australia.

For example, see the work by: Mola, J.M. and Williams, N.M., 2018. Fire‐induced change in floral abundance, density, and phenology benefits bumble bee foragers. Ecosphere9(1), p.e02056.

Line 92: “Our study will be” – your study *is* the first trailing. Speak in the affirmative – you have already conducted the study.

Line 152: Edit – “Bee hotels were be installed in August” – delete ‘be’

Figure 2A – needs some work. Caption sates “showing the number of median cavities occupied” however figure has both boxplots and points, so it is unclear what is actually being described. Also, y-axis is constrained such that whiskers of boxplots are cutoff, and y-axis scales are different between left and right plot. Need to extend y-axis to avoid this trimming of the whiskers and put both plots on the same y-axis scale. Also describe more precisely what data is being shown and consider what the overall purpose of this figure is. Because the trends over time are the same across sites (#ocuppied nests increases generally month to month – which is also being shown in figure 2B) I would suggest a simpler single plot showing the difference in # of occupied nest by site and method. In this case x-axis is would still be site but for each site there would be two boxplot bars (one for bamboo and one for wooden hotel – color coded). This setup would emphasize the fact that bamboo hotels had much higher occupancy.

Line 310: It is unclear how many hotels would produce the 1,200 native bees suggested here. Please clarify if this is the number of hotels used in the study – if so just remind the reader.

Line 313: “…there just under twice as many…” – missing word ‘were’

Line 317-18: ‘standaridsing’ should be standardise. Also no need to abbreviate ‘approximate’.

Line 403: Previous research is cited in this sentence but not cited formally in text as a reference. Please address.

Author Response

Bee hotels as a tool for post-fire recovery of cavity-nesting native bees Response to reviewer comments

Reviewer 1

Overall, this is a relevant study and one that addressing a growing concern related to climate change, its consequences (increases in wildfire), and the impacts on native bee populations. The study design is interesting, and the authors outline the pros and cons as well as potential pitfalls and future directions appropriately. Analytical methods are sufficient and clearly described. There are a few main considerations that should be addressed prior to publication. The first is a lack of reference to the broad series of literature of on the effects of fire on pollinator communities globally. Much work has been done in North America (and elsewhere) in both forest and grassland systems. There is room for this topic to be referenced however the authors choose in the introduction - especially the first paragraph (see specific comments below). The second main point of emphasis regards Figure 2A - this figure is a bit confusing (the caption related to 2A does not clearly describe what is shown) and I have provided some suggestions about how to make this figure clearer and more impactful. Lastly, I would suggest a careful read through the manuscript because I found several instances where either extra words were included in sentences or words were left out (some examples in specific comments below). 

 Response: thank you for your feedback on our ms and recommendations for improvement, which we have followed as detailed below.

Line 67: You state here: “Despite this, how native bee populations respond to fire, and practical solutions to aid in recolonization and recovery, have not been investigated. “

You already cite this reference in a previous sentence: Carbone, L.M., Tavella, J., Pausas, J.G. and Aguilar, R., 2019. A global synthesis of fire effects on pollinators. Global Ecology and Biogeography28(10), pp.1487-1498.

And that reference undermines your statement that native bee responses to fire has not been investigated.

There is a robust series of literature on native bee responses to fire in other systems globally that should be noted and can bolster the predictions presented here. I would suggest the inclusion of at least some citations of these studies and a reframing of this sentence if meant to state that this topic has not been studied in Australia.

For example, see the work by: Mola, J.M. and Williams, N.M., 2018. Fire‐induced change in floral abundance, density, and phenology benefits bumble bee foragers. Ecosphere9(1), p.e02056.

Response: We want to draw attention that practical solutions to aid in recolonisation and recovery have not been investigated, and we have now reframed the emphasise that the topic has not been studied in Australia.

I agree that there is literature overseas that would be good to cite in the introduction, and have now done so. (e.g. Mola & Williams, L59, and additional references, L.68)

Line 92: “Our study will be” – your study *is* the first trailing. Speak in the affirmative – you have already conducted the study.

Response: I have corrected this.

Line 152: Edit – “Bee hotels were be installed in August” – delete ‘be’

Response: I have corrected this.

Figure 2A – needs some work. Caption sates “showing the number of median cavities occupied” however figure has both boxplots and points, so it is unclear what is actually being described. Also, y-axis is constrained such that whiskers of boxplots are cutoff, and y-axis scales are different between left and right plot. Need to extend y-axis to avoid this trimming of the whiskers and put both plots on the same y-axis scale. Also describe more precisely what data is being shown and consider what the overall purpose of this figure is. Because the trends over time are the same across sites (#ocuppied nests increases generally month to month – which is also being shown in figure 2B) I would suggest a simpler single plot showing the difference in # of occupied nest by site and method. In this case x-axis is would still be site but for each site there would be two boxplot bars (one for bamboo and one for wooden hotel – color coded). This setup would emphasize the fact that bamboo hotels had much higher occupancy.

Response: thank you for this suggestion, I agree it would be better to display this as suggested, and I have redone the figure.

Line 310: It is unclear how many hotels would produce the 1,200 native bees suggested here. Please clarify if this is the number of hotels used in the study – if so just remind the reader.

Response: thank you I have revised this sentence.

Line 313: “…there just under twice as many…” – missing word ‘were’

Response: I have corrected.

Line 317-18: ‘standaridsing’ should be standardise. Also no need to abbreviate ‘approximate’.

Response: I have corrected.

Line 403: Previous research is cited in this sentence but not cited formally in text as a reference. Please address.

Response: I have corrected.

Reviewer 2 Report

Comments and Suggestions for Authors

Overall, I enjoyed reading this manuscript. The authors clearly put in a lot of time and effort and produced some novel and important data. Exploring how trap nests can be utilized as a conservation tool is an important area within bee conservation that has not been fully explored. Below are a few comments that I hope the authors find constructive. 

Not completely necessary but there are a handful of other papers that deal with 'trap nests'. Most of these came out of Florida and utilized similar trap nests that the author have used here. I think using these papers could help bolster some of the authors' findings. 

Apis mellifera- were these primarily from feral or managed colonies? If the latter, could there be management strategies implored between bee keepers and conservationists?

Aculeata (from Table 2)- Would a better taxonomic name be better? For example, Aculeata can refer to bees, wasps, and ants. You already have Eumeninae separated (vespids) and Isodontia (sphecids). Were the other wasps from these families or others?

Fewer bees were observed in areas without bee hotels and vice versa. I have no doubts that providing bee hotels increases bee activity. With that said, I’m curious as to how the authors could explain whether providing bee hotels may concentrate bees within areas (hence, bees are simply more spread out within controls) and not necessarily be increasing populations.  Might consider adding a statement to the discussion detailing this possibility.

Line 160- native bees should be ‘a native bee…’

Author Response

Reviewer 2 comments and response to comments:

Overall, I enjoyed reading this manuscript. The authors clearly put in a lot of time and effort and produced some novel and important data. Exploring how trap nests can be utilized as a conservation tool is an important area within bee conservation that has not been fully explored. Below are a few comments that I hope the authors find constructive. 

Response: thank you for you comments and feedback, we have made the suggested revisions are detailed below.

Not completely necessary but there are a handful of other papers that deal with 'trap nests'. Most of these came out of Florida and utilized similar trap nests that the author have used here. I think using these papers could help bolster some of the authors' findings. 

Response: we have now cited two such studies: L83, L389

Apis mellifera- were these primarily from feral or managed colonies? If the latter, could there be management strategies implored between bee keepers and conservationists?

Response: unfortunately it was hard to know if they were managed or feral colonies, as beekeepers place hives near the jarrah forest, but there are also many feral colonies. I agree however that recommendations on reducing this pressure should be included in the article, see L.412.

Aculeata (from Table 2)- Would a better taxonomic name be better? For example, Aculeata can refer to bees, wasps, and ants. You already have Eumeninae separated (vespids) and Isodontia (sphecids). Were the other wasps from these families or others?

Response: I agree – wasp (Aculeata here) were also eumenids, and the only other wasps were Isodontia. I have cleared this up, as well as the table as I feel it should be better presented as total occupancy at the end of the project , as this is what will be carried over to the next generation of insects to emerge.

Fewer bees were observed in areas without bee hotels and vice versa. I have no doubts that providing bee hotels increases bee activity. With that said, I’m curious as to how the authors could explain whether providing bee hotels may concentrate bees within areas (hence, bees are simply more spread out within controls) and not necessarily be increasing populations.  Might consider adding a statement to the discussion detailing this possibility.

Response: we have now included a statement about needing to conduct longer-terms studies to determine if these actions increase the populations over generations (L339-343)

Line 160- native bees should be ‘a native bee…’

Response: corrected.